SciPost Physics

# Nonadiabatic Nonlinear Optics and Quantum Geometry — Application to the Twisted Schwinger Effect

Shintaro Takayoshi[1,2], Jianda Wu[3], Takashi Oka[1,4,5*]

**1** Max Planck Institute for the Physics of Complex Systems, Dresden 01187, Germany
**2** Department of Physics, Konan University, Kobe 658-8501, Japan
**3** Tsung-Dao Lee Institute & School of Physics and Astronomy, Shanghai Jiao Tong University, Shanghai 200240, China
**4** Max Planck Institute for Chemical Physics of Solids, Dresden 01187, Germany
**5** The Institute for Solid State Physics, The University of Tokyo, Kashiwa, Chiba 277-8581, Japan * oka@issp.u-tokyo.ac.jp

September 10, 2021

## Abstract

We study the tunneling mechanism of nonlinear optical processes in solids induced by strong coherent laser fields. The theory is based on an extension of the Landau-Zener model with nonadiabatic geometric effects. In addition to the rectification effect known previously, we find two effects, namely perfect tunneling and counterdiabaticity at fast sweep speed. We apply this theory to the twisted Schwinger effect, *i.e.*, nonadiabatic pair production of particles by rotating electric fields, and find a nonperturbative generation mechanism of the opto-valley polarization and photo-current in Dirac and Weyl fermions.

# 1   Introduction

Today, geometric effects [1] in electron dynamics have become a central research topic in condensed matter [2]. In adiabatic processes, it is known that electrons acquiring a geometric phase provoke exotic effects such as quantum Hall effect [3, 4]. On the other hand, the importance of geometric effects in nonadiabatic processes have been overlooked except for a few examples such as the geometric amplitude factor [5–7] and counterdiabatic driving [8–10] as well as the modification of the adiabaticity condition [11, 12]. In an example of nonadiabatic dynamics governed by a time-dependent Hamiltonian, M.V. Berry showed that the tunneling probability can depend on the direction of the parameter sweep due to the geometric amplitude factor [5].

We revisit the problem of nonadiabatic geometric effects with a motivation to apply it to the twisted Schwinger effect in Dirac and Weyl Fermions. The Schwinger effect is fermion-antifermion pair production in strong electric fields [13–15] and is known to originate from nonadiabatic tunneling in the momentum space [16–20]. Previously, AC extensions of the Schwinger effect were studied for linearly polarized fields $E_x = E\cos(\Omega t)$ [21–23]. The results have common nature as the problems of strong-field ionization [24] and a particle escaping from an oscillating trap [25]. For low frequency, the tunneling is exponentially suppressed with a threshold known as the Schwinger limit [13–15]. For higher frequency (but still lower than the excitation gap), multiphoton excitation is activated and the excitation probability obeys a power law. Nonadiabatic geometric effects kicks in when we study pair production induced by rotating electric fields (or circularly polarized laser fields) $E_x + iE_y = Ee^{i\Omega t}$ [26, 27], which we coin as the "twisted Schwinger effect". If we assume momentum conservation, the problem of the twisted Schwinger effect can be recast to the Landau-Zener problem with a curved trajectory in the parameter space. This effective model is reminiscent of the twisted Landau-Zener model studied by M.V. Berry mentioned above [5]. We perform a numerical analysis of the effective model dynamics and find three geometric effects. The first is the sweep direction dependence, which we call rectification. This is the same phenomenon indicated in ref. [5] in terms of the geometric amplitude factor. The two other effects are perfect tunneling and counterdiabaticity at fast sweep speed. In order to clarify the origin of the effects, we "untwist" the model with a unitary transformation, and obtain the standard Landau-Zener model with an effective gap parameter depending on the geometric amplitude factor (see Eq.(8) below). We can understand the three nonadiabatic geometric effects in a unified way through the modulation of the effective gap. Recently, rectification in quantum tunneling has been studied in solid-state systems [28]. However, as far as we know, the perfect tunneling and counterdiabaticity at fast sweep has not been argued in previous studies.

In a condensed matter framework, a rotating electric field is created by a circularly polarized laser [29–31], or shaking an optical lattice [32], while in high energy physics, it mimics the field created by ions passing by each other in heavy-ion collision experiments [33]. The rotating electric fields are known to induce valley polarization [34, 35] and photo-currents in 2D and 3D Dirac/Weyl materials [36–38], respectively. Second order perturbation [39] has served as a theoretical framework to describe these phenomena.

Due to the development of strong coherent laser sources, an extension of the theory to the nonperturbative regime is being awaited. We show that the three nonadiabatic geometric effects, i.e., rectification, perfect tunneling and counterdiabaticity, play an important role in understanding the nonperturbative versions of the opto-valley polarization and photocurrents in 2D and 3D Dirac/Weyl materials which are microscopically caused by the twisted Schwinger effect. On the other hand, if these symmetries are broken, it is possible to realize finite $U(1)$ photocurrent in a similar way as in the optical absorption mechanism proposed in [38, 40, 41].

## 2  Nonadiabatic geometric effects in quantum tunneling

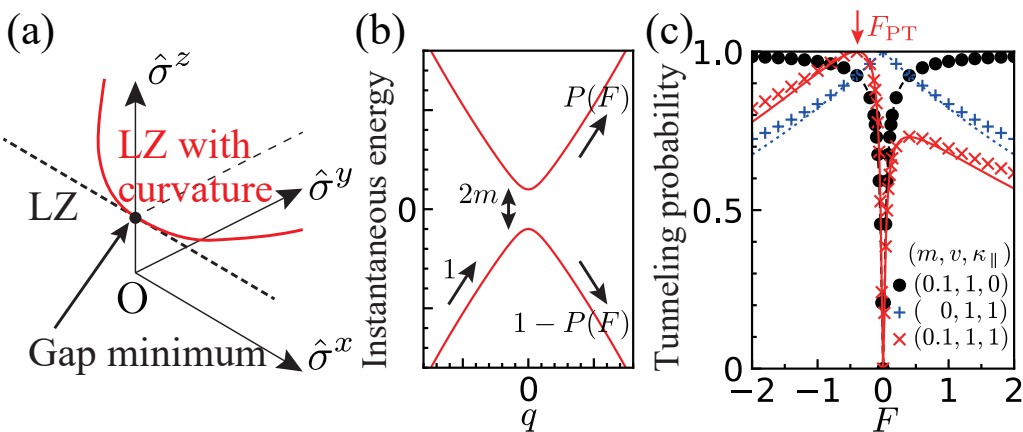

Figure 1: **Nonadiabatic geometric effects**: (a) Schematic picture of the LZ tunneling with curvature in parameter space. (b) Instantaneous energy of the Hamiltonian Eq. (1) with $(m, v, \kappa_\parallel) = (0.1, 1, 1)$ and schematic picture for quantum tunneling. (c) Tunneling probability $P(F)$ for Eq. (1) with a parameter sweep $q = -Ft$ obtained numerically (marks) compared with the tunneling formula Eq. (2) (lines).

We demonstrate the nonadiabatic geometric effects in a two level Hamiltonian with a parameter $q$ defined by

$$\hat{\mathcal{H}}(q) = m\hat{\sigma}^z + vq\hat{\sigma}^x + \frac{1}{2}\kappa_\parallel v^2 q^2 \hat{\sigma}^y, \tag{1}$$

where $\hat{\sigma}^j$ ($j = x, y, z$) is the Pauli matrix, $m$ is the gap, and $v$ ($> 0$) the energy slope. We use the unit $\hbar = c = 1$. The model Eq. (1) is a quadratic expansion of the twisted LZ model introduced by M. V. Berry [5]. We consider the diabatic tunneling problem in this Hamiltonian $\hat{\mathcal{H}}(q)$ which is formulated as follows.

1. We select an initial time $t = t_i (< 0)$. This is selected to be far enough from the time $t = 0$ when anti-crossing occurs.

2. Between the two eigenstates $\hat{\mathcal{H}}(q_i)|\pm(q_i)\rangle = E_\pm(q_i)|\pm(q_i)\rangle$ with $q_i = q(t_i)$, we select the initial state to be the lower energy eigenstate $|-(q_i)\rangle$.

3. The state evolves from the initial time $t = t_i$ to the final time $t = t_f \equiv |t_i|$ according to the Hamiltonian $\hat{\mathcal{H}}(q(t))$.

4. The solution is given as $|\psi(t_f)\rangle = \alpha|-(q_f)\rangle + \beta|+(q_f)\rangle$ where $|\pm(q_f)\rangle$ are the two eigenstates of $\hat{\mathcal{H}}(q_f)$ with $q_f = q(t_f)$. The tunneling probability is given by $P(F) = |\beta|^2$.

If we regard the coefficients of the Pauli matrices $\boldsymbol{x}(q) = (vq, \frac{1}{2}\kappa_{\parallel}v^2q^2, m)$ as a trajectory in the three-dimensional (3D) space, it defines a curve and $\kappa_{\parallel}$ is the geodesic curvature around the gap minimum in the parameter space [Fig. 1(a)]. The case of $\kappa_{\parallel} = 0$ corresponds to the Landau-Zener (LZ) Hamiltonian [16, 42]. The instantaneous energy of this Hamiltonian is plotted in Fig. 1(b). The tunneling probability $P(F)$ for a linear parameter sweep $q = -Ft$ in Eq. (1) can be evaluated and becomes (see subsection 2.1 for derivation)

$$P(F) = \exp\left[-\pi\frac{(m + \kappa_{\parallel}vF/4)^2}{v|F|}\right]. \tag{2}$$

Comparing this expression with the LZ formula, we notice that the effective tunneling gap is modified by the geodesic curvature.

The nonadiabatic geometric effects in the tunneling probability Eq.(2) can be related to the Berry connection and quantum geometry. Using the instantaneous eigenstates of the Hamiltonian satisfying

$$\hat{\mathcal{H}}(q)|\psi_m(q)\rangle = E_m(q)|\psi_m(q)\rangle, \tag{3}$$

with $m = \pm$, we define the Berry connection

$$\mathcal{A}_{nm}(q) = \langle\psi_n(t)|i\partial_q|\psi_m(t)\rangle. \tag{4}$$

The Berry connection relates the basis sets $|\psi_m(q)\rangle$ spanned by the instantaneous eigenstates at different parameter points $q$. We can define a gauge independent quantity

$$R_{nm}(q) = -A_{nn}(q) + A_{mm}(q) + \partial_q \arg A_{nm}(q) \tag{5}$$

known as the geometric amplitude factor [5] or the quantum geometric potential [11, 12]. In the Berry phase theory of polarization [43], where $q$ is regarded as the momentum in solids, $R_{nm}(q)$ is known as the shift vector that corresponds to the difference of the electric polarization between the $n$ and $m$-th bands [39]. In particular,

Quantum tunneling in the presence of the geometric amplitude factor has been studied [5, 11, 12, 28] and it was pointed out that this factor strongly affects the adiabaticity condition [11, 12]. We can see this by rewriting the tunneling probability using the geometric amplitude factor. The parameter $\kappa_{\parallel}$ in our quadratic Hamiltonian (1) is related to the geometric amplitude factor by

$$R_{+-}(q = 0) = v\kappa_{\parallel}, \tag{6}$$

and we can write the tunneling probability as ($\Delta = E_+ - E_-$ at $q = 0$)

$$P(F) = \exp\left[-\frac{\pi}{4v|F|}\left(\Delta + \frac{FR_{+-}}{2}\right)^2\right]. \tag{7}$$

This expression shows how quantum geometry affects the nonadiabatic tunneling process, where the effective tunneling gap is modified to

$$\Delta_{\text{eff}} = \Delta + \frac{FR_{+-}}{2}. \tag{8}$$

This expression reproduces the generalized adiabaticity condition obtained by one of the present authors in Refs. [11, 12].

The tunneling formula (7) predicts several interesting phenomena as we list below.

**Rectification** Although the instantaneous band structure is symmetric in $q \to -q$, the tunneling probability depends on the sign of $F$ and rectification happens [5]. The ratio $\gamma(F) \equiv P(|F|)/P(-|F|) = \exp\left(-\pi \frac{\Delta R_{+-}}{2v}\right)$ deviates from unity for $m \neq 0$ [Fig. 1(c)].

**Perfect tunneling** In conventional LZ tunneling, the tunneling probability monotonically increase from 0 (adiabatic) to 1 (diabatic limit or perfect tunneling) as the sweep speed increase. However, in the presence of nonadiabatic geometric effects, perfect tunneling is realized at finite sweep speed. For $m \neq 0$, $P(F)$ peaks out and becomes unity at a perfect tunneling sweeping speed $F_{\mathrm{PT}} = -2\Delta/R_{+-}$ indicated by an arrow in Fig. 1(c), which is determined from the condition $\Delta_{\mathrm{eff}} = 0$.

**Counterdiabaticity at fast sweep** For large $|F|$, $P(F)$ decreases as $\exp(-\pi R_{+-}^2 |F|/16v)$. In the extreme case of $m = 0$, the tunneling probability is a monotonically decreasing function of speed.

We have performed a numerical calculation of the tunneling probability using the Hamiltonian Eq. (1) and compared it with the tunneling formula Eq. (2) as depicted in Fig. 1(c). The results show good agreement and the above three nonadiabatic geometric effect is clearly seen.

For convenience, we also consider the two-band Hamiltonian with general operators up to $q^2$ order,

$$\mathcal{H} = \hat{A} + \hat{B}q + \hat{C}q^2/2. \tag{9}$$

The gap minimum and velocity extremum conditions at $q = 0$ require $\{\hat{A}, \hat{B}\} = 0$ and $\{\hat{B}, \hat{C}\} = 0$, respectively. This Hamiltonian is equivalent to the case of Eq. (11) with the parameters

$$m = \|\hat{A}\|, \quad v = \|\hat{B}\|, \quad \kappa_{\parallel}v^2 = -\frac{i}{8}\frac{\mathrm{Tr}\{[\hat{A}, \hat{B}], \hat{C}\}}{\|\hat{A}\|\|\hat{B}\|}, \tag{10}$$

where $\|\hat{O}\| \equiv \frac{1}{2}\sqrt{\mathrm{Tr}\{\hat{O}, \hat{O}\}}$.

## 2.1 A detailed derivation of the tunneling formula

In this subsection, we explain the derivation of the tunneling formula (Eq. (2)) for the Hamiltonian

$$\hat{\mathcal{H}}(q) = m\hat{\sigma}^z + vq\hat{\sigma}^x + \frac{1}{2}\kappa_{\parallel}v^2 q^2 \hat{\sigma}^y, \tag{11}$$

where $\hat{\sigma}^j$ ($j = x, y, z$) is the Pauli matrices, $m$ is the gap, $v$ the energy slope, and $\kappa_{\parallel}$ is the curvature around the gap minimum in the parameter space. The idea is to move to a local frame with trivial geometry, which we call the "LZ frame", and use the LZ formula or its extension: the Dykhne-Davis-Pechukas (DDP) (also known as the Landau-Dykhne or the imaginary time) method [44, 45] (see Ref. [20] for an extended discussion of the method).

Let us start from a general two-band Hamiltonian

$$\hat{\mathcal{H}}(q) = \boldsymbol{d}(q) \cdot \hat{\boldsymbol{\sigma}}, \tag{12}$$

where $\boldsymbol{d}(q)$ defines a curve in the Euclidean space. We consider tunneling at the gap minimum $q = 0$, and define the unit directional, tangential, and normal vectors as

$$\boldsymbol{r} = \boldsymbol{d}(0)/|\boldsymbol{d}(0)|$$
$$\boldsymbol{t} = \partial_q \boldsymbol{d}(0)/|\partial_q \boldsymbol{d}(0)|$$
$$\boldsymbol{n} = \boldsymbol{r} \times \boldsymbol{t}.$$

Note that $\boldsymbol{t} \perp \boldsymbol{r}$. We move to the LZ frame, where the curve $\boldsymbol{d}(q)$ is transformed to a curve on the plane spanned by $\boldsymbol{r}$ and $\boldsymbol{t}$ using a unitary operator $\hat{U} = e^{i\frac{\theta(q)}{2}\boldsymbol{r}\cdot\hat{\boldsymbol{\sigma}}}$. The angle $\theta(q)$ is determined as

$$\hat{U}^\dagger \hat{\mathcal{H}}(q) \hat{U} = [a(q)\boldsymbol{r} + b(q)\boldsymbol{t}] \cdot \hat{\boldsymbol{\sigma}},$$

where $a(q) = \boldsymbol{d}(q) \cdot \boldsymbol{r}$, $b(q) = \sqrt{|\boldsymbol{d}(q)|^2 - a(q)^2}$, and $\theta(q) = -\arctan\frac{\boldsymbol{d}(q)\cdot\boldsymbol{n}}{\boldsymbol{d}(q)\cdot\boldsymbol{t}}$. Then the Hamiltonian in the LZ frame becomes

$$\begin{aligned}\hat{\mathcal{H}}_{\mathrm{LZ}}(q) &= \hat{U}^\dagger \hat{\mathcal{H}}(q)\hat{U} - i\hat{U}^\dagger \partial_t \hat{U} \\ &= \left[\left(a(q) + \frac{\theta'(q)}{2}\frac{dq}{dt}\right)\boldsymbol{r} + b(q)\boldsymbol{t}\right] \cdot \hat{\boldsymbol{\sigma}}.\end{aligned} \tag{13}$$

In the case of the model Eq. (11), the parameters are $a(q) = m$, $b(q) = vq$, and $\theta'(q) = -\kappa_\parallel v/2$. Through the transformation, the additional quadratic term is eliminated and the gap is effectively modified from $m$ to $m_{\mathrm{eff}} = m + \kappa_\parallel vF/4$. The above formulation shows that the geometric meaning of $\kappa_\parallel$ is the curvature of $\boldsymbol{d}(q)$ in the plane spanned by $\boldsymbol{t}$ and $\boldsymbol{n}$ at $q = 0$.

With the application of the DDP method [44,45] for Eq. (13), the tunneling probability is expressed as

$$P \simeq \exp\left[-2\mathrm{Im}\int_0^{q_c} \frac{\Delta(q)}{|F(q)|}dq\right], \tag{14}$$

where $\Delta(q) = 2[(a(q) - \theta'(q)F(q)/2)^2 + b(q)^2]^{1/2}$ is the energy difference and $F(q) = -\frac{dq}{dt}$ is the Jacobian (expressed as function of $q$). In the DDP method, the integration path is deformed from the real axis, and the singular point closest to the real axis governs the tunneling probability. In Eq. (14), the integration is performed to $q_c$ (on the imaginary axis), which is defined as a point in complex plane where the gap vanishes $\Delta(q_c) = 0$ (the branching point of square root). For the linear sweep $q = -Ft$, the Jacobian is just $F(q) = -\frac{dq}{dt} = F$. Applying Eq. (14) to the model Eq. (11), and noticing $a(q) = m$, $b(q) = \sqrt{(vq)^2 + (\kappa_\parallel v^2 q^2/2)^2} = vq + \mathcal{O}(q^3)$, and $\theta'(q) = -\frac{d}{dq}\arctan(\kappa_\parallel vq/2) = -\kappa_\parallel v/2 + \mathcal{O}(q^2)$, we can calculate the tunneling probability as

$$\begin{aligned}P(F) &= \exp\left[-\frac{4}{|F|}\int_0^{\frac{1}{|v|}(m+\kappa_\parallel vF/4)}\sqrt{(m+\kappa_\parallel vF/4)^2 - (vq)^2}dq\right] \\ &= \exp\left[-\frac{\pi}{4}\frac{(2m+\kappa_\parallel vF/2)^2}{v|F|}\right]\end{aligned} \tag{15}$$

as given in Eq. (2).

# 3   Twisted Schwinger effect in 2D: Nonadiabatic opto-valleytronics

In the following sections, we study how nonadiabatic geometric effects in the tunneling probability Eq. (7) lead to nontrivial dynamics of electrons in Dirac and Weyl semimetals driven by strong electric laser fields.

We begin our analysis with the dynamics of 2D Dirac fermions in rotating electric fields. We introduce the field as gauge potential $\boldsymbol{A} = A(-\sin(\Omega t), \cos(\Omega t))$ [electric field $\boldsymbol{E} = E(\cos(\Omega t), \sin(\Omega t))$ $(E = A\Omega > 0)$], and the effective Hamiltonian for the fermions with chirality $\xi = \pm$ is given as

$$\hat{\mathcal{H}} = v[\xi(k_x + eA_x)\hat{\sigma}^x + (k_y + eA_y)\hat{\sigma}^y] + m\hat{\sigma}^z, \tag{16}$$

where $e$ ($> 0$) is the elementary charge, $v$ is the Fermi velocity, and $m$ ($> 0$) is the mass parameter. This model has implication to valleytronics in 2D materials such as monolayer transition metal dichalcogenide (TMD) and graphene [46, 47], where laser-induced valley polarization is demonstrated [34, 35, 48–52]. In these materials, the chirality $\xi$ corresponds to the valley index specifying the two Dirac points $K_\xi$ in the dispersion.

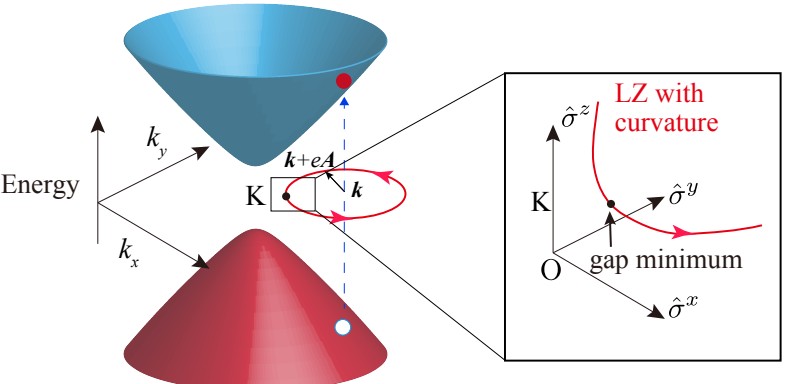

Figure 2: **Mapping from the twisted Schwinger effect to the twisted Landau Zener problem**: In rotating electric fields, the electron-hole pairs have a covariant momentum $\boldsymbol{k} + e\boldsymbol{A}(t)$ which performs a rotating motion in the momentum space. During this dynamics, the energy gap minimizes when $\boldsymbol{k} + e\boldsymbol{A}(t)$ is closest to the $K$-point. We focus on this gap minimum point as depicted in the right box. By performing a quadratic expansion of the Hamiltonian $\hat{\mathcal{H}}(t)$ in the time variable ($q = \Omega t$) around the gap minimum time, we obtain the twisted Landau Zener problem defined by Eq. (9).

We assume that the Fermi energy is zero, and the time evolution starts from a zero-temperature ground state. After the field is switched on at $t = 0$, nonadiabatic processes take place creating fermion-antifermion pairs. The tunneling process in momentum space can be mapped to a twisted Landau Zener problem discussed in the previous section as depicted in Fig. 2. We note that, in this mapping, we use a quadratic approximation. In this system, the laser frequency $\Omega$ plays the role of the speed parameter $F$ in the twisted LZ model. We allow $\Omega$ to be positive or negative which corresponds to the helicity specifying left or right circular polarization. The fermion-antifermion production probability per cycle of the laser field is given by

$$\mathcal{P}_\xi(\boldsymbol{k}) = \exp\left[-\pi \frac{\left(M - \frac{\xi\Omega m}{4M}\right)^2}{veE}\right], \tag{17}$$

where we defined $M = \sqrt{v^2(|\boldsymbol{k}| - eE/(|\Omega|))^2 + m^2}$. To derive this expression, we have expanded the Hamiltonian (16) around the time that minimizes the energy gap up to quadratic order obtaining the form (10) and used the tunneling formula Eq. (2). We note that the remaining analysis is based on this approximate treatment (quadratic expansion) and the results are not exact.

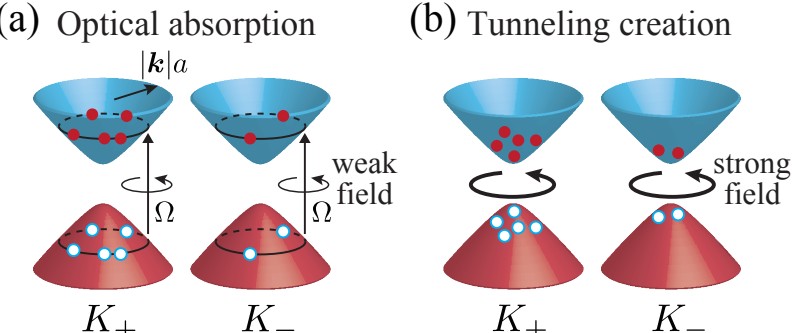

Figure 3: **Perturbative v.s. Nonadiabatic opto-valley polarization in 2D gapped Dirac fermion**: Schematic picture of pair excitations at the two valleys through two mechanisms. (a) In optical absorption, the pairs are concentrated on an equal energy curve $\Delta E(\boldsymbol{k}) = \Omega$ due to energy conservation. (b) In tunneling creation, the pairs are produced according to Eq. (17) (see Fig. 4).

In Fig. 3, we schematically compare the pair production induced by (a) standard optical absorption process, and by (a) tunneling with nonadiabatic geometric effects.

**(a) Perturbative optical absorption process** In the case of standard optical absorption process, a perturbative picture of optical absorption is employed, where we consider the eigenstates $|\psi_n(\boldsymbol{k})\rangle$ of the single body Hamiltonian $\hat{\mathcal{H}}(\boldsymbol{k})$ satisifying $\hat{\mathcal{H}}(\boldsymbol{k})|\psi_n(\boldsymbol{k})\rangle = E_n|\psi_n(\boldsymbol{k})\rangle$. Electrons in the occupied bands are excited to the unoccupied bands, and the energy difference of the electron and hole is given by the photon energy. The momentum dependence of the excitation density is determined by the optical selection rule encoded in the transition dipole moment. In electrons in solids, the optical transition between bands $m$ and $n$ ($m \neq n$) is given by the perturbation $\sum_{j=x,y,z} E^j(t)\mathcal{A}^j_{mn}(\boldsymbol{k})$ to the Hamiltonian $\hat{\mathcal{H}}(\boldsymbol{k})$. The transition dipole moment is given by the Berry connection $\mathcal{A}^j_{mn}(\boldsymbol{k}) = \langle\psi_m|i\partial_j|\psi_n\rangle$ ($j = x, y, z$) [39]. The photo absorption rate of circularly polarized laser in the $K_\xi$ valley becomes $PA_{\xi=\pm} \propto |\mathcal{A}^\xi_{mn}(\boldsymbol{k})|^2|E(\Omega)|^2\delta(\Delta E - \Omega)$, where we defined $\mathcal{A}^{\xi=\pm}_{mn}(\boldsymbol{k}) = \mathcal{A}^x_{mn}(\boldsymbol{k}) \pm i\mathcal{A}^y_{mn}(\boldsymbol{k})$ and $\Delta E = E_c - E_v$ [34,35].

**(b) Nonadiabatic optical absorption process** In the case of tunneling excitations, the properties of the excited pairs are different from the perturbative case. Energy is no longer conserved since the Hamiltonian Eq. (16) is depends on time. Electron and hole pairs can be created even when their energy difference is not equal to the photon energy. The role of the photo absorption rate $PA_{\xi=\pm}$ is now played by the production probability $\mathcal{P}_\xi(\boldsymbol{k})$ given in Eq. (17) or more generically in Eq. (7).

We summarize the comparison in Table 1.

## 3.1 Valley polarization via tunneling creation

In Fig. 4, we plot the production probability for several $\Omega$. We see that there is a strong chirality dependence, and the sign of $\xi\Omega$ determines whether excitations are "optically allowed" ($\xi\Omega > 0$) or "optically forbidden" ($\xi\Omega < 0$). This difference originates from the geometric amplitude factor. In this sense, the optical selection rule [34,35] in perturbative optics is replaced by the nonadiabatic geometric effects when nonperturbative strong field

Table 1: **Perturbative v.s. Nonadiabatic opto-valley polarization**

| Perturbative nonlinear optics [34, 35] | Nonadiabatic nonlinear optics (this work) |
| --- | --- |
| Optical absorption | Tunneling creation of electron-hole pairs |
| $PA_\xi \propto \vert \mathcal{A}_{mn}^\xi(\boldsymbol{k})\vert^2 \vert E(\Omega)\vert^2 \delta(\Delta E - \Omega)$ | Tunneling probability $\mathcal{P}_\xi(\boldsymbol{k})$ |
| Optical selection rule (transition dipole) | Geometric amplitude factor |
| Energy momentum conservation modulo photon | Non-conservation of energy and momentum |
| Valley polarization $\gamma = \frac{PA_+(\boldsymbol{k})}{PA_-(\boldsymbol{k})}$ | Valley polarization $\gamma = \frac{\mathcal{P}_+(\boldsymbol{k})}{\mathcal{P}_-(\boldsymbol{k})}$ |

excitations are considered. The ratio of the production rates between the two chiralities

$$\gamma = \frac{\mathcal{P}_+(\boldsymbol{k})}{\mathcal{P}_-(\boldsymbol{k})} = \exp\left(\frac{\pi \Omega m}{veE}\right) \tag{18}$$

is independent of the wavenumber. In the gapless case, as in graphene, $\gamma$ is unity and there is no valley dependence. When the gap parameter $m$ is finite, as in monolayer TMD, imbalance becomes finite and the ratio exponentially grows or decays with increasing $\vert\Omega\vert/E$.

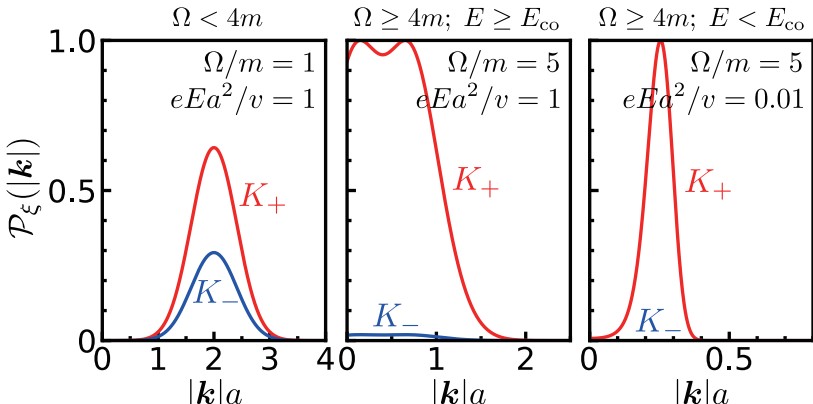

Figure 4: **Tunneling probability in 2D gapped Dirac fermion**: The wavenumber dependence of the production probability $\mathcal{P}_\xi(\boldsymbol{k})$. The parameters are $(\Omega/m, eEa^2/v) = (1,1), (5,1), (5,0.01)$ and $ma/v = 0.5$, where $a$ is the lattice constant.

Next, let us study the wavenumber dependence of the production probability as depicted in Fig. 4. The distribution is rotationally symmetric and only depends on $\vert\boldsymbol{k}\vert a$ ($a$: lattice constant). They have peaks as shown in Fig. 4 at $\vert\boldsymbol{k}\vert = k_{\text{peak}}$, where

$$k_{\text{peak}} = \begin{cases} \dfrac{eE}{\vert\Omega\vert} & (\Omega < 4m), \tag{19} \\[2ex] \dfrac{eE}{\vert\Omega\vert} \pm \dfrac{1}{v}\sqrt{m(\xi\Omega/4 - m)} & (\Omega \geq 4m;\ E \geq E_{\text{co}}), \tag{20} \\[2ex] \dfrac{eE}{\vert\Omega\vert} + \dfrac{1}{v}\sqrt{m(\xi\Omega/4 - m)} & (\Omega \geq 4m;\ E < E_{\text{co}}). \tag{21} \end{cases}$$

The crossover field is defined by

$$E_{\text{co}} = \frac{\vert\Omega\vert}{ev}\sqrt{m(\xi\Omega/4 - m)}. \tag{22}$$

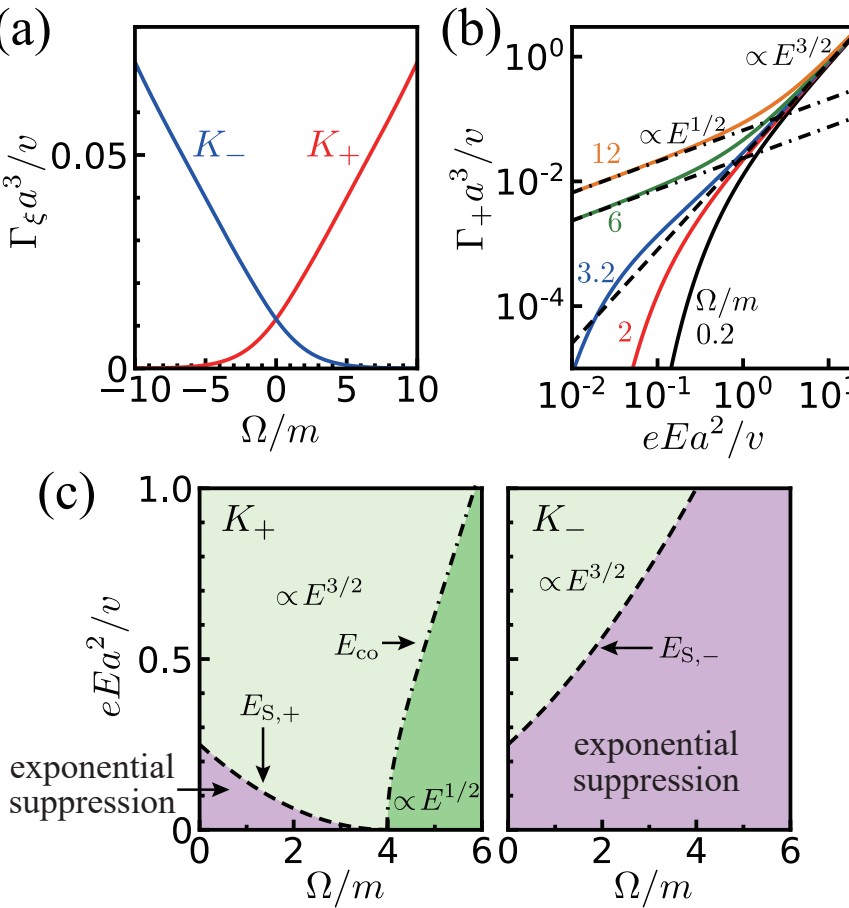

Figure 5: **2D gapped Dirac fermion**: (a) The total pair production rate per unit of time and volume. We fix $ma/v = 0.5$ and $eEa^2/v = 1$. (b) The electric field dependence of the total production rate $\mathcal{P}_\xi^{\text{tot}}$. (c) $(E, \Omega)$-phase diagram of the twisted Schwinger effect.

We can understand the peak structure from the wavenumber dependent effective mass parameter in Eq. (17) defined by

$$m_{\text{eff}} = M - \xi\Omega m/(4M). \tag{23}$$

The peaks are dictated by the wavenumber minimizing the effective mass and their properties qualitatively change depending on whether the frequency $\Omega$ is below or above $4m$. For $\Omega < 4m$, the distributions have a single peak at the wavenumber where $m_{\text{eff}} > 0$ is minimized. On the other hand, for higher frequencies $\Omega \geq 4m$, perfect tunneling takes place at the optically allowed valley ($\xi\Omega > 0$) when the effective gap $m_{\text{eff}}$ close. There is a crossover when the electric field is increased, The number of perfect tunneling peaks changes from one for $E < E_{\text{co}}$ to two for $E \geq E_{\text{co}}$. This field strength $E_{\text{co}}$ characterizes a crossover of the total production rate which we will explain below.

## 3.2 Crossover in the production rate

We define the total fermion-antifermion production rate per unit of time and volume as $\Gamma_\xi \equiv \frac{|\Omega|}{(2\pi)^3} \int d\boldsymbol{k} \mathcal{P}_\xi(\boldsymbol{k})$ and plot it against frequency in Fig. 5(a). We see clearly the rectification effect where the imbalance ratio $\Gamma_+/\Gamma_- = \gamma$ increases exponentially for large

$\Omega/E$ following Eq. (18). In the low-frequency region, it takes the form (Appendix A)

$$\Gamma_\xi \simeq \frac{eE}{(2\pi)^2}\sqrt{\frac{eE}{v}}\exp\left(-\pi\frac{E_{\mathrm{S},\xi}}{E}\right). \tag{24}$$

Here we define the Schwinger limit of field strength as

$$E_{\mathrm{S},\xi} \equiv (\overline{m}_{\mathrm{eff},\xi})^2/(ve) = (m-\xi\Omega/4)^2/(ve), \tag{25}$$

where $\overline{m}_{\mathrm{eff},\xi}$ is the effective mass at the gap minimizing wavenumber Eq. (19). Equation (24) is an extension of Schwinger's production rate evaluated originally for a DC electric field to the case of rotating electric field. For $\Omega = 0$, Eq. (24) coincides with the 2D version of Schwinger's result [13, 19] with the QED Schwinger limit $E_{\mathrm{S}} = m_e^2 c^3/(\hbar e)$ obtained by replacing $m \to m_e c^2$ and $v \to \hbar c$.

Figure 5(b) shows the electric field dependence of the production rate with the optically allowed chirality ($\xi\Omega > 0$) for several frequencies. For strong fields, all curves converge to the dashed line $\Gamma_\xi \to \frac{eE}{(2\pi)^2}\sqrt{\frac{eE}{v}}$ described by the asymptotic form of Eq. (24) independent of $\Omega$. For weak fields, we observe two different behaviors. The low frequency ($\Omega < 4m$) curves drop below the dashed line following Eq. (24) due to the exponential suppression of tunneling at weak fields. In contrast, curves for high frequency ($\Omega \geq 4m$) turn above the dashed line and converge to a $\Gamma_+ \propto E^{1/2}$ behavior. In Fig. 5(c), we summarize the tunneling behaviors into a $(E, \Omega)$-phase diagram, which we explain below.

**Low frequency ($|\Omega| < 4m$) (Appendix A.1)**   The Schwinger limit $E = E_{\mathrm{S},\xi}$ [Eq. (25)] characterizes the crossover from the weak field exponentially suppressed regime to the $\Gamma_\xi \propto E^{3/2}$ behavior at strong field. Increasing $\Omega$ from zero, the Schwinger limit $E_{\mathrm{S},\xi}$ for the optically allowed chirality ($\xi\Omega > 0$) decreases and becomes zero at $|\Omega| = 4m$, where perfect tunneling starts to happen. In contrast, for the optically forbidden chirality ($\xi\Omega < 0$), $E_{\mathrm{S},\xi}$ monotonically increase against $|\Omega|$. This suppression of tunneling is the consequence of counterdiabaticity in the twisted LZ tunneling.

**High frequency ($|\Omega| \geq 4m$) (Appendix A.2)**   For the optically allowed chirality $\xi\Omega > 0$, the effective gap closes and the Schwinger limit vanishes due to perfect tunneling. There is a crossover taking place around $E = E_{\mathrm{co}}$ defined in Eq. (22) where the number of the peaks in the distribution function changes (Fig. 4). The production rate shows the $\Gamma_\xi \propto E^{3/2}$ behavior at strong field $E > E_{\mathrm{co}}$ and changes to a $\Gamma_\xi \propto E^{1/2}$ behavior at weak fields $E < E_{\mathrm{co}}$. In particular, in the weak field regime, the production rate shows an asymptotic form

$$\Gamma_\xi \simeq \frac{|\Omega|}{(4\pi)^2 v}\sqrt{|\Omega|m}\sqrt{\frac{eE}{v}} \tag{26}$$

for optically allowed $\xi$, which is evaluated in the appendixA.2. On the other hand, for the optically forbidden chirality $\xi\Omega < 0$, the Schwinger limit monotonically increases as $|\Omega|$ increases.

Before closing this section, we give an estimate of the Schwinger limit in solid-state materials. The Fermi velocity $v$ and mass parameters $m = \Delta/2$ for a typical TMD material $MoS_2$ is given by $v = 3.5$ ÅeV and $m = \frac{\Delta}{2} = 0.83$ eV where $\Delta$ is the optical gap [35]. The Schwinger limit of $MoS_2$ is given by $E_{\mathrm{S},\xi}(0) = m^2/(ve) = 0.20$ V/Å $= 2.0 \times 10^9$ V/m for $\Omega = 0$. For finite photon energy $\Omega$, the Schwinger limit decreases and vanish at $\Omega = 4m = 2\Delta = 3.3$ eV for one valley. Experimentally realizable fields using THz laser

$(\Omega \sim 0)$ is around $E_{\text{THz}} = 10^8$ V/m [53], while it exceeds $E_{\text{NI}} = 10^9$ V/m [54] in the near-infrared region $\Omega = 0.6 - 1$ eV. Thus, the field strength of near-infrared lasers is comparable to the Schwinger limit at finite photon energy $\Omega$ and can be used to verify our predictions.

## 4 Twisted Schwinger effect in 3D: Nonadiabatic photo-current

Next, we proceed to an analysis of 3D massless Dirac fermions subject to rotating electric fields described by the Hamiltonian

$$
\begin{aligned}
\hat{\mathcal{H}}_{\text{3D}} =& v \sum_{j=x,y,z} \hat{\gamma}^0 \hat{\gamma}^j (q_j + eA_j) \\
=& \begin{pmatrix} -v\sum_{j=x,y,z}(q_j + eA_j)\hat{\sigma}^j & 0 \\ 0 & v\sum_{j=x,y,z}(q_j + eA_j)\hat{\sigma}^j \end{pmatrix}
\end{aligned}
\tag{27}
$$

with the 3D wave number $\boldsymbol{q} = (\boldsymbol{k}, k_z)$, $\boldsymbol{A} = A(-\sin(\Omega t), \cos(\Omega t), 0)$ and the gamma matrices

$$
\hat{\gamma}^0 = \begin{pmatrix} 0 & I \\ I & 0 \end{pmatrix}, \quad \hat{\gamma}^j = \begin{pmatrix} 0 & \hat{\sigma}^j \\ -\hat{\sigma}^j & 0 \end{pmatrix} \; (j = x, y, z).
$$

We can recast this Hamiltonian to the 2D Dirac Hamiltonian studied in the previous section. By performing the unitary transform

$$
\hat{U} = \begin{pmatrix} \exp(i\frac{\pi}{2}\hat{\sigma}^x) & 0 \\ 0 & I \end{pmatrix}
$$

to the Hamiltonian Eq. (27), we obtain

$$
\hat{U}^\dagger \hat{\mathcal{H}}_{\text{3D}} \hat{U} = \begin{pmatrix} \hat{\mathcal{H}}_- & 0 \\ 0 & \hat{\mathcal{H}}_+ \end{pmatrix},
$$

where

$$
\hat{\mathcal{H}}_\xi = v[\xi(k_x - eA\sin(\Omega t))\hat{\sigma}^x + (k_y + eA\cos(\Omega t))\hat{\sigma}^y + k_z\hat{\sigma}^z]
\tag{28}
$$

is the Weyl Hamiltonian with chirality $\xi = \pm$. This Hamiltonian is equivalent to the 2D Dirac Hamiltonian Eq. (16) studied in the previous section with the replacement of the mass $m$ by $vk_z$. Thus, the fermion-antifermion production probability per cycle of the laser field is given by

$$
\mathcal{P}_\xi(\boldsymbol{k}) = \exp\left[ -\pi \frac{\left(M - \frac{\xi\Omega v k_z}{4M}\right)^2}{veE} \right],
\tag{29}
$$

where we defined $M = v\sqrt{(|\boldsymbol{k}| - eE/|\Omega|)^2 + k_z^2}$.

Below, we assume that the Fermi energy is at the Dirac point and exploit the scaling symmetry rewriting the model with variables $\tilde{t} = |\Omega|t$ and $\tilde{\boldsymbol{q}} = v\boldsymbol{q}/|\Omega|$. The Schrödinger equation is recast to $i\partial_{\tilde{t}}|\Psi_\xi(\tilde{t})\rangle = \hat{\tilde{\mathcal{H}}}_\xi|\Psi_\xi(\tilde{t})\rangle$ with $\hat{\tilde{\mathcal{H}}}_\xi = \xi(\tilde{k}_x - \text{sgn}(\Omega)\tilde{A}\sin\tilde{t})\hat{\sigma}^x + (\tilde{k}_y + \tilde{A}\cos\tilde{t})\hat{\sigma}^y + \tilde{k}_z\hat{\sigma}^z$. Then we can set the frequency $|\Omega|$ to unity and

$$
\tilde{A} = veA/|\Omega| = veE/\Omega^2
\tag{30}
$$

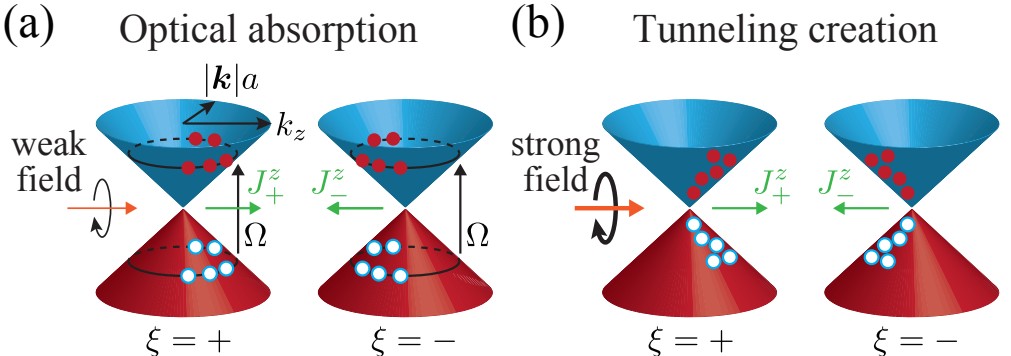

Figure 6: **Nonadiabatic v.s. Perturbative photo-current generation in 3D massless Dirac fermion**: Schematics of pair production and resulting photocurrent in the two Weyl components induced by the (a) tunneling creation and (b) optical excitations. (a) In tunneling creation, the pairs are produced according to Eq. (29). See Fig. 7 for the results. (b) In optical absorption, the pairs are concentrated on equal energy curves due to energy conservation.

is the unique scaling parameter that characterizes the field strength.

In Fig. 6, we schematically compare the pair production in the 3D Dirac systems induced by (a) standard optical absorption process to (b) tunneling creation. Similarly to the 2D case summarized in Fig. 3 and Table 1, the nonadiabatic geometric effects take the place of the optical selection rule [38]. Circularly polarized laser field propagating along the $z$ axis induces vertical transitions that are imbalanced between $\pm k_z$. The imbalance in the created pairs result in a photocurrent $J_\xi^z$ for each Weyl component $\xi = \pm$ as we will see below. In Dirac systems with chiral and mirror reflection symmetries, the total photocurrent cancels since the production rate $\mathcal{P}_\xi(\tilde{\boldsymbol{q}})$ is symmetric under $\xi \to -\xi$, $k_z \to -k_z$. On the other hand, if these symmetries are broken, it is possible to realize finite $U(1)$ photocurrent in a similar way as in the optical absorption mechanism proposed in [38, 40, 41].

## 4.1 Expression of the total and chiral current

The total U(1) and chiral current operators are represented as

$$\hat{J}^z = -ve\hat{U}^\dagger\hat{\gamma}^0\hat{\gamma}^z\hat{U} = -ve\begin{pmatrix} \hat{\sigma}^z & 0 \\ 0 & \hat{\sigma}^z \end{pmatrix},$$

and

$$\hat{J}_5^z = -ve\hat{U}^\dagger\hat{\gamma}_5\hat{\gamma}^0\hat{\gamma}^z\hat{U} = -ve\begin{pmatrix} -\hat{\sigma}^z & 0 \\ 0 & \hat{\sigma}^z \end{pmatrix},$$

where $\hat{\gamma}_5 = \begin{pmatrix} -I & 0 \\ 0 & I \end{pmatrix}$. To evaluate the expectation value of the currents, we need to estimate the distribution of the electron-hole pairs. This can be done by calculating the time evolution of the density matrix in the presence of relaxation. One of the schemes is to employ the Liouville von Neumann equation in the momentum space within the relaxation time approximation [55]. However, for simplicity, here we assume that the system is on-shell, i.e., the density matrix is diagonal in the eigenstate basis and the distribution is obtained by the balance between the creation process characterized by the

tunneling probability and the relaxation time. This can be done by first representing the density matrix as $\rho_{\boldsymbol{q},\xi}(t) = n_{\boldsymbol{q},\xi}(t)|\Psi_{\boldsymbol{q},\xi,1}\rangle\langle\Psi_{\boldsymbol{q},\xi,1}| + [1 - n_{\boldsymbol{q},\xi}(t)]|\Psi_{\boldsymbol{q},\xi,2}\rangle\langle\Psi_{\boldsymbol{q},\xi,2}|$, where $|\Psi_{\boldsymbol{q},\xi,1}\rangle$ and $|\Psi_{\boldsymbol{q},\xi,2}\rangle$ are the states for upper and lower bands with chirality $\xi$. The master equation is

$$\frac{dn_{\boldsymbol{q},\xi}(t)}{dt} = [1 - n_{\boldsymbol{q},\xi}(t)]\mathcal{P}_\xi(\boldsymbol{q})\frac{|\Omega|}{2\pi} - n_{\boldsymbol{q},\xi}(t)/\tau, \tag{31}$$

where $\tau$ is the relaxation time. Note that we have the factor $\frac{|\Omega|}{2\pi}$ (= inverse of the time period) in the first term on the r.h.s. since $\mathcal{P}_\xi(\boldsymbol{q})$ is defined as the tunneling probability per cycle. In the steady state $dn_{\boldsymbol{q},\xi}(t)/dt = 0$, if we assume that the relaxation time is short $|\Omega|\tau\mathcal{P}_\xi(\boldsymbol{q})/(2\pi) \ll 1$, we obtain $n_{\boldsymbol{q},\xi}(t) = |\Omega|\tau\mathcal{P}_\xi(\boldsymbol{q})/(2\pi)$. The current density for the component of chirality $\xi$ is provided as

$$J_\xi^z = -2ve\frac{|\Omega|\tau}{2\pi a^3}\left(\frac{a}{2\pi}\right)^3\int d\boldsymbol{q}\frac{k_z}{\sqrt{|\boldsymbol{k}|^2 + k_z^2}}\mathcal{P}_\xi(\boldsymbol{q}) = -\frac{2e\tau|\Omega|^4}{(2\pi)^4v^2}\int d\tilde{\boldsymbol{q}}\frac{\tilde{k}_z}{\sqrt{|\tilde{\boldsymbol{k}}|^2 + \tilde{k}_z^2}}\mathcal{P}_\xi(\tilde{\boldsymbol{q}}). \tag{32}$$

We can calculate the total and chiral (spin) currents as $J^z = J_+^z + J_-^z$ and $J_5^z = J_+^z - J_-^z$.

We also define total (chiral) production rates as $\Gamma_{\text{tot}}^{3D} = \Gamma_+^{3D} + \Gamma_-^{3D}$ ($\Gamma_5^{3D} = \Gamma_+^{3D} - \Gamma_-^{3D}$) using $\Gamma_\xi^{3D} = \frac{|\Omega|^4}{(2\pi)^4v^3}\int d\tilde{\boldsymbol{q}}\mathcal{P}_\xi(\tilde{\boldsymbol{q}})$. Due to the symmetry of $\mathcal{P}_\xi(\tilde{\boldsymbol{q}})$ under $\xi \to -\xi$, $k_z \to -k_z$, $\Gamma_5^{3D} = J^z = 0$ holds.

## 4.2 Novel crossover between weak-to-strong field behaviors

Now, let us discuss the physical consequence of the geometric nonadiabatic effect in the tunneling creation in 3D Dirac fermions. In the massless Dirac and Weyl fermions, there is no tunneling threshold and we expect that the total production rate shows a power-law behavior against the electric field strength. We show that there is a crossover between the weak and strong field regimes accompanied by a change in power.

In Figs. 7(a)-7(c), we plot the production probability $\mathcal{P}_\xi(\tilde{\boldsymbol{q}})$ for $\xi = +$ and $\Omega > 0$ obtained in Eq. (29). It is rotationally symmetric around the $\tilde{k}_z$ axis. The production probability for the other chirality $\xi = -$ is a reflection of $\xi = +$ around the $\tilde{k}_z = 0$ plane. The production probability shows peaks around the wavenumber satisfying the perfect tunneling conditions Eqs. (20) and (21) with $m$ replace by $k_z$. In the plane of $(|\tilde{\boldsymbol{k}}|, \tilde{k}_z)$, the perfect tunneling peaks define a circle centered at $(|\tilde{\boldsymbol{k}}|, \tilde{k}_z) = (\tilde{A}, 1/8)$ with a radius $1/8$ and are plotted as black solid curves. We find a crossover in the shape of the perfect tunneling peaks that occurs at

$$\tilde{A}_{\text{co}} = 1/8. \tag{33}$$

For $\tilde{A} < \tilde{A}_{\text{co}}$ the circle is incomplete and approaches a semicircle in the small $\tilde{A}$ limit, and for large field $\tilde{A} \geq \tilde{A}_{\text{co}}$ the circle becomes complete. Remembering the definition of $\tilde{A}$ given in Eq. (30), the crossover field strength is $E_{\text{co}} = \frac{1}{8}\frac{(\hbar\Omega)^2}{e\hbar v}$, where we have temporally recovered the Planck constant. For example, in the case of $Cd_3As_2$, the velocity parameter is of the order of $v \sim 10^5 \text{m/s}$ and using $\hbar = 6.6 \times 10^{-16}\text{eVs}$, the crossover fields for photon energies $\hbar\Omega = 1\text{eV}$ and $\hbar\Omega = 1\text{meV}$ are $E_{\text{co}} \sim 2 \times 10^9 \text{V/m}$ and $E_{\text{co}} \sim 2 \times 10^3 \text{V/m}$, respectively. We stress that these parameters for the laser strength are experimentally feasible.

Next, we investigate how this crossover is seen in the physically observable quantities. We plot the total production rate $\Gamma_\xi^{3D}$ and the chiral current in the $z$ direction $J_5^z$ in Fig. 7(d). The quantities show a power-law behavior in the weak and strong field limits

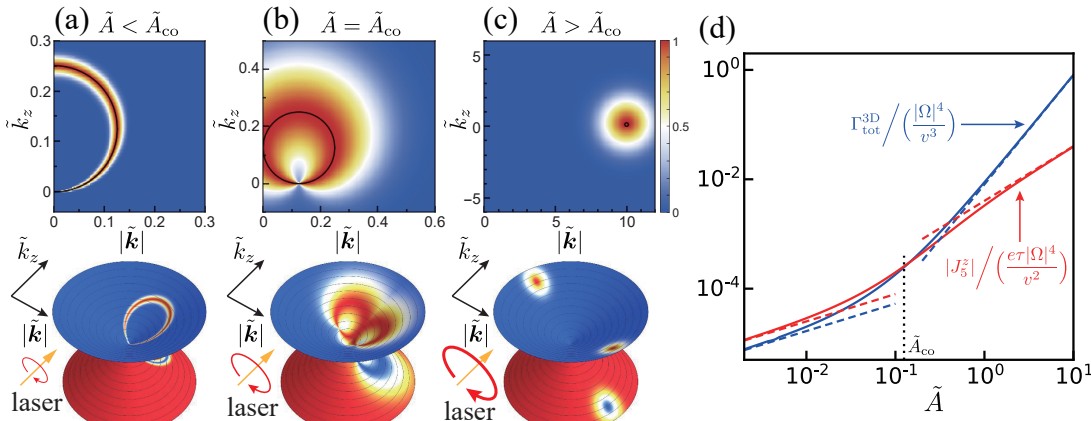

Figure 7: **3D massless Dirac fermion**: (a)-(c) The production probability $\mathcal{P}_\xi(\tilde{\boldsymbol{q}})$ for chirality $\xi = +$ plotted for several field strength parameters (a) $\tilde{A} = veE/\Omega^2 = 0.001$, (b) $\tilde{A} = 1/8$, and (c) $\tilde{A} = 10$. They are rotationally symmetric around the $\tilde{k}_z$ axis and the probability for particles with chirality $\xi = -$ is reflected as $\tilde{k}_z \to -\tilde{k}_z$. The solid black curve denotes wavenumber at which perfect tunneling occurs [Eqs. (20) and (21) with $m$ replace by $k_z$]. The lower panels show the fermion-antifermion pairs on the Weyl cone $E = \pm\sqrt{|\tilde{\boldsymbol{q}}|}$ for fixed $\tilde{k}_y = 0$. (d) The total production rate and chiral current are plotted as blue and red solid curves while the dashed lines represent their asymptotic power-law behavior Eqs. (34) and (35).

with different powers. The change of the power occurs around the crossover field $\tilde{A} = \tilde{A}_{\mathrm{co}}$ and their asymptotic behaviors are given by

$$\Gamma_{\mathrm{tot}}^{\mathrm{3D}} \Big/ \left(\frac{|\Omega|^4}{v^3}\right) \to \begin{cases} \frac{1}{3(4\pi)^3}\tilde{A}^{1/2} & (\tilde{A}/\tilde{A}_{\mathrm{co}} \ll 1) \\ \frac{2}{(2\pi)^3}\tilde{A}^2 & (\tilde{A}/\tilde{A}_{\mathrm{co}} \gg 1), \end{cases} \tag{34}$$

$$J_5^z \Big/ \left(\frac{-e\tau|\Omega|^4}{v^2}\right) \to \begin{cases} \frac{\mathrm{sgn}(\Omega)}{2(4\pi)^3}\tilde{A}^{1/2} & (\tilde{A}/\tilde{A}_{\mathrm{co}} \ll 1) \\ \frac{\mathrm{sgn}(\Omega)}{(2\pi)^3}\tilde{A}^1 & (\tilde{A}/\tilde{A}_{\mathrm{co}} \gg 1). \end{cases} \tag{35}$$

It is possible to analytically evaluate the asymptotic behaviors using the fact that the distribution around the peak is a Gaussian with a width scaling as $\tilde{A}^{1/2}$. The detailed calculation is given in appendix B.

This is a novel nonperturbative crossover that originated from the the nonadiabatic geometric effect that has no perturbative analogue. Let us discuss how we can measure the current as well as the crossover in solid-state experiments. We have discussed a general theory based on Weyl and Dirac Hamiltonians. There are various material realizations of Weyl and Dirac Hamiltonians [56], where the chirality $\xi$ may correspond to degrees of freedom such as orbitals and spins as well as their mixtures. For example, in $Co_3Sn_2S_2$ [57] the chirality $\xi$ corresponds to spin [58] and the chiral current $J_5^z$ can be detected as a spin current. The generation of $U(1)$ photocurrent due to optical absorption in Weyl semimetals with broken symmetry have been studied in refs. [37,38,40,41,59]. If the Fermi energy is non-zero and the system has finite carrier density, the nonlinear anomalous Hall current [60] can also contribute to the current generation in the $z$-direction [61]. The three mechanisms, i.e., tunneling creation, optical absorption, nonperturbative Hall current, have different dependencies on the laser and material parameters such as Field strength, photon energy, and Fermi energy. The asymptotic behaviors of the physical observables in Eq. (34), (35) is useful in identifying the origin of the photoinduced current.

# 5 Conclusion

We studied the nonadiabatic geometric effects in quantum tunneling and found that they provoke anomalous phenomena such as rectification, perfect tunneling and counterdiabaticity. We derived the tunneling formula describing these effects through the modulation of the effective mass. We studied the implication of nonadiabatic geometric effects in the Schwinger effect, i.e., tunneling creation of carriers, induced by rotating electric fields. Two condensed matter applications are mentioned. One is the valley polarization that can be induced in 2D Dirac materials, and the other is the generation of spin (and charge) current in 3D Dirac (and Weyl) materials. Our finding adds another example to the rich nonperturbative phenomena induced by circularly polarized laser in electronic systems [62–68]. Finally, we comment that the interplay between the nonadiabatic geometric effects and interaction is an open problem calling for further study. We point out that there is an interesting resemblance between the phase diagram of the twisted Schwinger effect [Fig. 5(c)] and that of a strongly interacting holographic model [69, 70].

# Acknowledgements

We would like to thank Masamitsu Hayashi, Ryo Shimano, Sota Kitamura, Takahiro Morimoto, Masafumi Udagawa, Francesco Peronaci, Alexandra Landsman, Hamed Koochaki Kelardeh, and Lisa Ortmann for fruitful discussions.

**Funding information** This work was supported by JSPS KAKENHI Grant No. JP21K03412 and JST CREST Grant No. JPMJCR19T3, Japan. J. W. acknowledges additional support from a Shanghai talent program. The work at Shanghai Jiao Tong University is sponsored by Natural Science Foundation of Shanghai with Grant No. 20ZR1428400 and Shanghai Pujiang Program with Grant No. 20PJ1408100 (JW)

# A Detailed calculations for the 2D Dirac fermions

We consider the Hamiltonian

$$\hat{\mathcal{H}} = v[\xi(k_x - eA\sin q)\hat{\sigma}^x + (k_y + eA\cos q)\hat{\sigma}^y] + m\hat{\sigma}^z.$$

where $q = \Omega t$. It is expanded as to $q$ up to the second order and can be written in the form of Eq. (9) with

$$\begin{aligned}
\hat{A} =& m\hat{\sigma}^z + \xi v k_x \hat{\sigma}^x + v(k_y + eA)\hat{\sigma}^y \\
\hat{B} =& -\xi v eA\hat{\sigma}^x \\
\hat{C} =& -veA\hat{\sigma}^y.
\end{aligned}$$

Let us consider the tunneling at $k_x = 0$, $k_y < 0$ for $\Omega > 0$ and $k_x = 0$, $k_y > 0$ for $\Omega < 0$ (i.e., $k_y = -\mathrm{sgn}(\Omega)|\boldsymbol{k}|$) in the time interval of $-\pi/|\Omega| \le t \le \pi/|\Omega|$. Note that the sign of $A$ is the same as that of $\Omega$. The parameters in Eq. (11) are given as

$$\begin{aligned}
m \to& \sqrt{v^2(-\mathrm{sgn}(\Omega)|\boldsymbol{k}| + eA)^2 + m^2} \\
v \to& veA \\
\kappa_\parallel v^2 \to& \frac{\xi m veA}{\sqrt{v^2(-\mathrm{sgn}(\Omega)|\boldsymbol{k}| + eA)^2 + m^2}}.
\end{aligned}$$

Since $F = -\Omega$ and $E = A\Omega$, the tunneling probability for twisted Schwinger effect in 2D Dirac fermions is given as

$$\mathcal{P}_\xi(\boldsymbol{k}) = \exp\left[-\pi\frac{\left(M - \frac{\xi\Omega m}{4M}\right)^2}{veE}\right],\tag{36}$$

where we defined

$$M = \sqrt{v^2(|\boldsymbol{k}| - eE/|\Omega|)^2 + m^2}.$$

We investigate the total probability

$$\mathcal{P}_\xi^{\text{tot}} \equiv \left(\frac{a}{2\pi}\right)^2 \int d\boldsymbol{k}\,\mathcal{P}_\xi(\boldsymbol{k}) = \frac{a^2}{2\pi}\int_0^\infty dk\,k\,\mathcal{P}_\xi(k).$$

below focusing on the case of $\xi\Omega > 0$.

## A.1  Low frequency region

When the laser frequency is smaller than double the gap $|\Omega| < 4m$, $\mathcal{P}_\xi$ shows a peak at $|k| = eE/|\Omega|$ in the momentum space. Let expand Eq. (36) around $k = eE/|\Omega|$. We represent $k' = k - eE/|\Omega|$, and since $vk' \ll m$, we can approximate as $(m/M)^2 = (1 + v^2 k'^2/m^2)^{-1} \simeq 1 - v^2 k'^2/m^2$. Hence, in the low frequency region,

$$\mathcal{P}_\xi(\boldsymbol{k}) \simeq \exp\left[-\frac{\pi}{veE}\left(m^2 + v^2 k'^2 - \frac{\xi\Omega m}{2} + \frac{\Omega^2}{16}(1 - v^2 k'^2/m^2)\right)\right]$$

$$= \exp\left[-\frac{\pi}{veE}\left\{v^2\left(1 - \frac{\Omega^2}{16m^2}\right)k'^2 + \left(m - \frac{\xi\Omega}{4}\right)^2\right\}\right],\tag{37}$$

which is the normal distribution with the standard deviation $\sqrt{eE/(2\pi v)}(1 - \Omega^2/(16m^2))^{-1/2}$. When $\sqrt{eE/(2\pi v)}(1 - \Omega^2/(16m^2))^{-1/2} \ll eE/(|\Omega|)$, $\mathcal{P}_\xi^{\text{tot}}$ can be calculated as

$$\mathcal{P}_\xi^{\text{tot}} \simeq \exp\left[-\frac{\pi}{veE}\left\{v^2\left(1 - \frac{\Omega^2}{16m^2}\right)k'^2 + \left(m - \frac{\xi\Omega}{4}\right)^2\right\}\right]$$

$$= \frac{eEa^2}{2\pi|\Omega|}\sqrt{\frac{eE}{v}}\left(1 - \frac{\Omega^2}{16m^2}\right)^{-1/2}\exp\left[-\frac{\pi}{veE}\left(m - \frac{\xi\Omega}{4}\right)^2\right].$$

Therefore the e-h production rate per unit of time and volume is provided as

$$\Gamma_\xi \equiv \frac{|\Omega|}{2\pi a^2}\mathcal{P}_\xi^{\text{tot}} \simeq \frac{eE}{(2\pi)^2}\sqrt{\frac{eE}{v}}\exp\left[-\frac{\pi}{veE}\left(m - \frac{\xi\Omega}{4}\right)^2\right].\tag{38}$$

## A.2  High frequency region

In the high frequency region $|\Omega| > 4m$, $\mathcal{P}_\xi(\boldsymbol{k})$ have peaks at the perfect tunneling points

$$k = \frac{eE}{|\Omega|} \pm \frac{1}{v}\sqrt{m(\xi\Omega/4 - m)}$$

instead of $k = eE/|\Omega|$. In the case of strong electric field, however, the broadening of $\mathcal{P}_\xi(\boldsymbol{k})$ is much larger than the distance between the perfect tunneling points $\sqrt{eE/(2\pi v)} \gg \sqrt{m(\xi\Omega/4 - m)}/v$ and the contribution to $\mathcal{P}_\xi^{\text{tot}}$ mainly comes from $k < eE/|\Omega| - \sqrt{m(\xi\Omega/4 - m)}/v$ and $k > eE/|\Omega| + \sqrt{m(\xi\Omega/4 - m)}/v$, where the approximation Eq. (37) is still valid. Hence the e-h production rate per unit of time and volume is provided by Eq. (38).

In the case of weak electric field, the contribution to $\mathcal{P}_\xi^{\text{tot}}$ comes from the wavenumber around the perfect tunneling point $k = eE/(|\Omega|) + \sqrt{m(\xi\Omega/4 - m)}/v$. By expanding $\mathcal{P}_\xi(\boldsymbol{k})$ around this wave number, i.e., $k = eE/|\Omega| + \sqrt{m(\xi\Omega/4 - m)}/v + k'$, we obtain

$$
\begin{aligned}
\mathcal{P}_\xi(\boldsymbol{k}) &\simeq \exp\left[ -\xi\frac{4\pi}{veE\Omega m}\left(vk'\sqrt{m(\xi\Omega - 4m)} + v^2 k'^2\right)^2\right] \\
&\simeq \exp\left[ -\xi\frac{4\pi v}{eE\Omega}(\xi\Omega - 4m)k'^2\right].
\end{aligned}
\tag{39}
$$

Thus, by noting $eE/|\Omega| \ll \sqrt{m(\xi\Omega/4 - m)}/v$, the total production rate is given as

$$
\mathcal{P}_\xi^{\text{tot}} \simeq \frac{a^2}{8\pi v}\sqrt{\xi\Omega m}\sqrt{\frac{eE}{v}}.
\tag{40}
$$

Thus the e-h production rate per unit of time and volume is given as

$$
\Gamma_\xi = \frac{|\Omega|}{(4\pi)^2 v}\sqrt{\xi\Omega m}\sqrt{\frac{eE}{v}}.
\tag{41}
$$

## B  Detailed calculations for the 3D Dirac fermions

In the same way as the 2D case, the tunneling probability for twisted Schwinger effect in 3D Dirac fermions is given as

$$
\mathcal{P}_\xi(\tilde{\boldsymbol{q}}) = \exp\left[ -\pi\frac{\left(M - \dfrac{\xi\text{sgn}(\Omega)\tilde{k}_z}{4M}\right)^2}{\tilde{A}}\right],
\tag{42}
$$

with

$$
M = \sqrt{(|\tilde{\boldsymbol{k}}| - \tilde{A})^2 + \tilde{k}_z^2}.
$$

Then the e-h production rate per unit time and volume for each chirality is given as

$$
\Gamma_\xi^{3D} = \frac{|\Omega|}{2\pi a^3}\left(\frac{a}{2\pi}\right)^3\int d\boldsymbol{q}\,\mathcal{P}_\xi(\boldsymbol{q}) = \frac{|\Omega|^4}{(2\pi)^4 v^3}\int d\tilde{\boldsymbol{q}}\,\mathcal{P}_\xi(\tilde{\boldsymbol{q}}).
\tag{43}
$$

We can calculate the total and chiral e-h production rates as $\Gamma_{\text{tot}}^{3D} = \Gamma_+^{3D} + \Gamma_-^{3D}$ and $\Gamma_5^{3D} = \Gamma_+^{3D} - \Gamma_-^{3D}$.

The main contribution to the production rates and currents come from the wavenumbers around the perfect tunneling points

$$
|\tilde{\boldsymbol{k}}| = \tilde{A} \pm \sqrt{\tilde{k}_z(\xi\text{sgn}(\Omega)/4 - \tilde{k}_z)} \quad (\xi\text{sgn}(\Omega)/8 - 1/8 \le \tilde{k}_z \le \xi\text{sgn}(\Omega)/8 + 1/8).
\tag{44}
$$

Equation (44) is rewritten as

$$
(|\tilde{\boldsymbol{k}}| - \tilde{A})^2 + (\tilde{k}_z - \xi\text{sgn}(\Omega)/8)^2 = (1/8)^2,
$$

which forms a circle with the center $(\tilde{A}, \xi\text{sgn}(\Omega)/8)$ and the radius $1/8$ or a part of it in the $|\boldsymbol{k}|$-$k_z$ space.

## B.1  Weak field regime

In the weak field regime $\tilde{A} \ll 1/8$, the contribution mainly comes from the positive sign branch of Eq. (44) $|\tilde{\boldsymbol{k}}| = \tilde{A} + \sqrt{\tilde{k}_z(\xi\mathrm{sgn}(\Omega)/4 - \tilde{k}_z)}$. The tunneling probability can be approximated as

$$\mathcal{P}_\xi(\tilde{\boldsymbol{q}}) \simeq \exp\Big[ - \xi\mathrm{sgn}(\Omega)\frac{4\pi}{\tilde{A}}(\xi\mathrm{sgn}(\Omega) - 4\tilde{k}_z)\tilde{k}'^2\Big], \tag{45}$$

where $\tilde{k}' = |\tilde{\boldsymbol{k}}| - \tilde{A} - \sqrt{\tilde{k}_z(\xi\mathrm{sgn}(\Omega)/4 - \tilde{k}_z)}$. Then we can calculate the production rate Eq. (43) as

$$
\begin{aligned}
&\Gamma_\xi^{3D}\Big/\Big(\frac{|\Omega|^4}{v^3}\Big) \\
&\simeq \frac{1}{(2\pi)^4} \int d\tilde{\boldsymbol{q}}\exp\Big[ - \xi\mathrm{sgn}(\Omega)\frac{4\pi}{\tilde{A}}(\xi\mathrm{sgn}(\Omega) - 4\tilde{k}_z)\tilde{k}'^2\Big] \\
&= \frac{1}{2(2\pi)^3}\int_{\xi\mathrm{sgn}(\Omega)/8-1/8}^{\xi\mathrm{sgn}(\Omega)/8+1/8} d\tilde{k}_z\Big(\tilde{A} + \sqrt{\tilde{k}_z(\xi\mathrm{sgn}(\Omega)/4 - \tilde{k}_z)}\Big)\sqrt{\frac{\tilde{A}}{\xi\mathrm{sgn}(\Omega)(\xi\mathrm{sgn}(\Omega) - 4\tilde{k}_z)}} \\
&\simeq \frac{1}{4(2\pi)^3}\int_{\xi\mathrm{sgn}(\Omega)/8-1/8}^{\xi\mathrm{sgn}(\Omega)/8+1/8} d\tilde{k}_z\sqrt{\xi\mathrm{sgn}(\Omega)\tilde{A}\tilde{k}_z} = \frac{A^{1/2}}{6(4\pi)^3}.
\end{aligned}
\tag{46}
$$

Therefore

$$\Gamma_{\mathrm{tot}}^{3D}\Big/\Big(\frac{|\Omega|^4}{v^3}\Big) = \frac{A^{1/2}}{3(4\pi)^3}, \quad \Gamma_5^{3D}\Big/\Big(\frac{|\Omega|^4}{v^3}\Big) = 0. \tag{47}$$

For the calculation of currents, noting that

$$|\tilde{\boldsymbol{k}}|^2 + \tilde{k}_z^2 \simeq \Big(\tilde{A} + \sqrt{\tilde{k}_z(\xi\mathrm{sgn}(\Omega)/4 - \tilde{k}_z)}\Big)^2 + \tilde{k}_z^2 \simeq \xi\mathrm{sgn}(\Omega)\tilde{k}_z/4,$$

we can derive

$$
\begin{aligned}
J_\xi^z\Big/\Big(\frac{-e\tau|\Omega|^4}{v^2}\Big) &\simeq \frac{4}{(2\pi)^4}\int d\tilde{\boldsymbol{q}}\,\xi\mathrm{sgn}(\Omega)\sqrt{\xi\mathrm{sgn}(\Omega)\tilde{k}_z}\mathcal{P}_\xi(\tilde{\boldsymbol{q}}) \\
&\simeq \frac{1}{(2\pi)^3}\int_{\xi\mathrm{sgn}(\Omega)/8-1/8}^{\xi\mathrm{sgn}(\Omega)/8+1/8} d\tilde{k}_z\tilde{A}^{1/2}\tilde{k}_z = \xi\mathrm{sgn}(\Omega)\frac{\tilde{A}^{1/2}}{4(4\pi)^3}
\end{aligned}
\tag{48}
$$

Therefore

$$J^z\Big/\Big(\frac{-e\tau|\Omega|^4}{v^2}\Big) = 0, \quad J_5^z\Big/\Big(\frac{-e\tau|\Omega|^4}{v^2}\Big) = \mathrm{sgn}(\Omega)\frac{\tilde{A}^{1/2}}{2(4\pi)^3}. \tag{49}$$

## B.2  Strong field regime

In the strong field regime $\tilde{A} \gg 1/8$, the contribution mainly comes from both sign branches of Eq. (44). The tunneling probability can be approximated as

$$\mathcal{P}_\xi(\tilde{\boldsymbol{q}}) \simeq \exp\Big[ - \frac{\pi}{\tilde{A}}\Big\{\tilde{k}'^2 + \Big(\tilde{k}_z - \frac{\xi\mathrm{sgn}(\Omega)}{4}\Big)^2\Big\}\Big], \tag{50}$$

where $\tilde{k}' = |\tilde{\boldsymbol{k}}| - \tilde{A}$. Then we can calculate the production rate Eq. (43) as

$$\Gamma_\xi^{3D} \Big/ \Big(\frac{|\Omega|^4}{v^3}\Big) \simeq \frac{1}{(2\pi)^4} \int d\tilde{\boldsymbol{q}} \exp\Big[ -\frac{\pi}{\tilde{A}}\Big\{\tilde{k}'^2 + \Big(\tilde{k}_z - \frac{\xi\,\text{sgn}(\Omega)}{4}\Big)^2\Big\}\Big]$$

$$= \frac{\tilde{A}^{3/2}}{(2\pi)^3} \int_{-\infty}^{\infty} d\tilde{k}_z \exp\Big[ -\frac{\pi}{\tilde{A}}\Big(\tilde{k}_z - \frac{\xi\,\text{sgn}(\Omega)}{4}\Big)^2\Big] = \frac{A^2}{(2\pi)^3}. \tag{51}$$

Therefore

$$\Gamma_{\text{tot}}^{3D} \Big/ \Big(\frac{|\Omega|^4}{v^3}\Big) = \frac{2A^2}{(2\pi)^3}, \quad \Gamma_5^{3D} \Big/ \Big(\frac{|\Omega|^4}{v^3}\Big) = 0. \tag{52}$$

For the calculation of currents, noting that $|\tilde{\boldsymbol{k}}|^2 + \tilde{k}_z^2 \simeq \tilde{A}^2$, we can derive

$$J_\xi^z \Big/ \Big(\frac{-e\tau|\Omega|^4}{v^2}\Big) \simeq \frac{2}{(2\pi)^4\tilde{A}} \int d\tilde{\boldsymbol{q}}\,\tilde{k}_z \mathcal{P}_\xi(\tilde{\boldsymbol{q}})$$

$$\simeq \frac{2\tilde{A}^{1/2}}{(2\pi)^3} \int_{-\infty}^{\infty} d\tilde{k}_z\,\tilde{k}_z \exp\Big[ -\frac{\pi}{\tilde{A}}\Big(\tilde{k}_z - \frac{\xi\,\text{sgn}(\Omega)}{4}\Big)^2\Big] = \xi\,\text{sgn}(\Omega)\frac{\tilde{A}}{2(2\pi)^3} \tag{53}$$

Therefore

$$J^z \Big/ \Big(\frac{-e\tau|\Omega|^4}{v^2}\Big) = 0, \quad J_5^z \Big/ \Big(\frac{-e\tau|\Omega|^4}{v^2}\Big) = \text{sgn}(\Omega)\frac{\tilde{A}}{(2\pi)^3}. \tag{54}$$

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
