# Peer review of "Nonadiabatic Nonlinear Optics and Quantum Geometry --- Application to the Twisted Schwinger Effect"

_SciPost Physics_

## Round 2 · Referee Report · Michael Sentef (Referee 1) · 2021-6-21

Strengths

1- This paper identifies a new aspect of Bloch function quantum geometry, namely strong corrections to the valley polarization and valley currents in the presence of nonlinear terms in a Dirac Hamiltonian. A simple and transparent derivation is provided, with a new formula for the Landau-Zener tunneling problem (eq. 2, the main result of this work).

2- Predictions are made for 2D and 3D systems, which can be tested experimentally (e.g., discussion on page 14).

Weaknesses

1- Maybe a minor weakness is that the paper does not mention and discuss the role of dissipation and decoherence in the quantum tunneling process, which is known to be relevant in light-driven Dirac systems, and can be modelled by a relatively simple extension of their model(s). See, e.g., the relevance of dissipation and decoherence discussed in a theory work related to the anomalous Hall effect measurement in driven graphene, https://journals.aps.org/prb/abstract/10.1103/PhysRevB.99.214302

Report

Overall, I think that the paper clearly meets the acceptance criteria for SciPost Physics. This is an original theoretical proposal which highlights the importance of quantum geometric effects for strong-field phenomena in general and specifically for the Landau-Zener problem, which opens a new pathway in an existing or a new research direction, with clear potential for multipronged follow-up work. (Criterion #3 for Acceptance in SciPost Physics). All of the necessary criteria are fulfilled.

Requested changes

1- I would like to see a discussion of the expected impact of dissipation and decoherence effects, as outlined above; I believe that such effects could be the study of a follow-up work.

---

## Round 2 · Referee Report · Anonymous (Referee 2) · 2021-7-13

Report

The Authors study the problem of a fast sweep of an external parameter across an energy barrier. The probability of particle tunneling is calculated. Their calculations are based on a simple two-band model. Applications to the case of driven 2D Dirac fermions and 3D Weyl fermions with circularly polarized light are presented. The main difference with respect to the usual Landau-Zener problem is that the parameter space is curved. They found interesting effects namely, asymmetrical tunneling probability with respect to direction of drive, perfect tunneling at finite sweep rate, and exponentially decreasing tunneling probability for large sweep rate. I find the paper results worth of publication in SciPost Physics. I however have few comments that should be addressed before publication.

  1. The Tunneling probability in Eq. 10 should be defined.

  2. Comment about v \kappa_{\parallel} `corresponding to the shift vector’ should be either fully explained or removed. Page 5.

  3. Why should the expansion of Eq.11 in powers of \Omega to second order be equivalent to the model Eq. 4 on which the results of this paper as based? E.g., why cubic terms in expansion of Eq. 11 are not important near the band minimum?

  4. Give an order of magnitude estimate of the value of the electric field and \Omega need to observe these effects for a realistic condensed matter realization.

---

## Round 2 · Author Response

Dear Editor,

Thank you very much for your careful reading and important comments.

We are grateful for giving us the opportunity to clarify the following point.

The "twisted" Landau-Zener model refers to a Hamiltonian with a linear dispersion in one direction and a parabolic dispersion in the perpendicular direction. This is not applicable to the Dirac or Weyl fermions one typically encounters in graphene or Weyl semimetals (although there do exist more complex models where this might apply).

We are afraid that there might be a misunderstanding in the interpretation of our Hamiltonian (1) that defines the twisted Landau-Zener problem. Our target systems are indeed the Dirac and Weyl fermions encountered in graphene or Weyl semimetals and NOT the semi-Dirac materials (= linear + quadratic band touching). The Hamiltonian (1) for the twisted Landau-Zener model is a quadratic time-dependent Hamiltonian, which can be realized in many different situations. The situation we consider is realized when a circularly polarized laser field is applied to graphene or Weyl semimetals as is described in Hamiltonian (11). This Hamiltonian (11) can be mapped to the twisted Landau Zener problem (1) (and its generalization (4)). In order to clarify this point, we added Figure 2 “Mapping from the twisted Schwinger effect to the twisted Landau Zener problem” to the manuscript.

As a computational exercise it may well be publishable, but for SciPost Physics the physical significance of the analysis is an essential ingredient.

The problem of Dirac and Weyl semimetals in a circularly polarized laser field has physical significance, which has led to important experiments such as Y. H. Wang, H. Steinberg, P. Jarillo-Herrero, and N. Gedik, “Observation of Floquet-Bloch states on the surface of a topological insulator”, Science 342, 453 (2013) and J. W. McIver, B. Schulte, F.-U. Stein, T. Matsuyama, G. Jotzu, G. Meier and A. Cavalleri, “Light-induced anomalous Hall effect in graphene”, Nat. Phys. 16(1), 38 (2020). Our work is expected to shed new light on these experiments.

Thus, we believe that our paper is suited for SciPost Physics rather than SciPost Physics Core.

---

## Round 2 · List of Changes

We added Figure 2.

---

## Round 3 · Author Response

[Reply to the Anonymous Report 1] We would like to deeply thank the referee for reading the manuscript carefully and giving us a positive report. Let us comment on the dissipation and decoherence effect indicated by the referee. Indeed, most of our analysis has been done within the framework of pure quantum evolution without the effect of the environment. The exception is the calculation of the photo-current given in (32), where we have used the steady-state solution of the master equation (31) which phenomenologically incorporates the dissipation effect. A more detailed analysis based on the Liouville von Neumann equation as performed in the reference mentioned by the referee (https://journals.aps.org/prb/abstract/10.1103/PhysRevB.99.214302) would be an interesting future project. We have added this comment in the main text above eq. (31).

[Reply to the Anonymous Report 2] We would like to deeply thank the referee for reading the manuscript carefully and giving us useful comments. We have modified the manuscript accordingly.

  1. "The Tunneling probability in Eq. 10 should be defined. " We have added a paragraph describing how we define the tunneling probability below (1).

  2. "Comment about v \kappa_{\parallel} `corresponding to the shift vector’ should be either fully explained or removed. Page 5." We thank the referee for pointing this out. The shift vector is an important quantity in the quantum geometric approach to non-linear optics. In order to fully explain this aspect in a self-contained way, we have performed several modifications to the manuscript. [1] We added two paragraphs on page 4 to give the definition and explanation of the shift vector (which is equivalent to the geometric amplitude factor in tunneling theory). The expression Eq. (7) gives the tunneling formula with the geometric amplitude factor. [2] Since quantum geometry is an important aspect of this work, we have changed the title to “Nonadiabatic Nonlinear Optics and Quantum Geometry — Application to Twisted Schwinger Effect” to stress its prominent role. We also changed the section titles (2. and 3.) to fit well with the change of the title. [3] We also added table I and its explanation on page 9 to fully explain the role of quantum geometry and the shift vector in non-linear optics. This gives a contrast between the perturbative theories of optical processes and the nonperturbative tunneling theory (which includes the shift vector contribution). We would like to acknowledge the referee for giving us the opportunity to make these changes, which we think will improve the value of this manuscript.

  3. "Why should the expansion of Eq.11 in powers of \Omega to second order be equivalent to the model Eq. 4 on which the results of this paper as based? E.g., why cubic terms in expansion of Eq. 11 are not important near the band minimum?" The concern of the referee is indeed correct. This is the limitation of our work where all the results are obtained within a quadratic approximation. We did not try to go to further order since the derivation of the tunneling formula (2) and (7) was already based on a quadratic approximation (see text above (15)). Thus, it does not make sense to keep higher-order terms of the Hamiltonian while using the tunneling formula. The limitation of the quadratic approximation is already seen in Fig. 1 (c) where the numerical results are compared with the tunneling formula. Although they coincide well in the week field (=slow speed) regime, the deviation increase as the field (speed) becomes stronger (faster). We agree that a more detailed comparison with numerical calculation is necessary. However, we would like to leave this for future study.

  4. "Give an order of magnitude estimate of the value of the electric field and \Omega need to observe these effects for a realistic condensed matter realization." For the 2D case, we have added a paragraph on page 11 at the end of section 3, where we estimate the Schwinger limit in a TMD material. It is reachable with current laser fields. For the 3D case, we already have an estimate of the crossover field below (33). This is also achievable with current laser fields.

---

## Round 3 · List of Changes

1. We have added a comment on the role of dissipation and decoherence effect in the main text above eq. (31).
2. We have added a paragraph describing how we define the tunneling probability below (1).
3. In order to make the explanation on the shift vector more self-contained, and highlight the importance of the concept of quantum geometry, we have made the following modifications. [1] We added two paragraphs in page 4 to give the definition and explanation of the shift vector (which is equivalent to the geometric amplitude factor in tunneling theory). The expression Eq. (7) gives the tunneling formula with the geometric amplitude factor. [2] Since quantum geometry is an important aspect of this work, we have changed the title to “Nonadiabatic Nonlinear Optics and Quantum Geometry — Application to Twisted Schwinger Effect” to stress its prominent role. We also changed the section titles (2. and 3.) to fit well with the change of the title. [3] We also added table I and its explanation in page 9 to fully explain the role of quantum geometry and the shift vector in non-linear optics. This gives a contrast between the perturbative theories of optical processes and the nonperturbative tunneling theory (which includes the shift vector contribution).
We added remarks on experimental feasibility in page 11 at the end of section 3.

You are currently on this page

Resubmission scipost_202104_00022v3 on 10 September 2021

---

## Editorial Decision

publication_decision_taken:_accept